# Practical Bayesian Algorithm Execution via Posterior Sampling

**Chu Xin Cheng**[*]
California Institute of Technology
ccheng2@caltech.edu

**Raul Astudillo**[*]
California Institute of Technology
rastudil@caltech.edu

**Thomas Desautels**
Lawrence Livermore National Laboratory
desautels2@llnl.gov

**Yisong Yue**
California Institute of Technology
yyue@caltech.edu

## Abstract

We consider Bayesian algorithm execution (BAX), a framework for efficiently selecting evaluation points of an expensive function to infer a property of interest encoded as the output of a base algorithm. Since the base algorithm typically requires more evaluations than are feasible, it cannot be directly applied. Instead, BAX methods sequentially select evaluation points using a probabilistic numerical approach. Current BAX methods use expected information gain to guide this selection. However, this approach is computationally intensive. Observing that, in many tasks, the property of interest corresponds to a target set of points defined by the function, we introduce *PS-BAX*, a simple, effective, and scalable BAX method based on posterior sampling. PS-BAX is applicable to a wide range of problems, including many optimization variants and level set estimation. Experiments across diverse tasks demonstrate that PS-BAX performs competitively with existing baselines while being significantly faster, simpler to implement, and easily parallelizable, setting a strong baseline for future research. Additionally, we establish conditions under which PS-BAX is asymptotically convergent, offering new insights into posterior sampling as an algorithm design paradigm.

## 1 Introduction

Many real-world problems can be cast as estimating a property of a black-box function with expensive evaluations. Bayesian optimization (BO) [1] has focused on the case where the property of interest is the function's global optimum. Similarly, level set estimation [2] deals with the problem of estimating the subset of points above (or below) a user-specified threshold.

In many cases, an algorithm to compute the property of interest is available, which we refer to as the *base algorithm*. However, this algorithm typically requires more evaluations than are feasible in practice and cannot be used directly. Instead, evaluation points must be carefully selected through other means. Similar to BO and level set estimation, the Bayesian algorithm execution (BAX) framework selects evaluation points using two key components: (1) a Bayesian probabilistic model of the function and (2) a sequential sampling criterion that leverages this model to choose new points for evaluation [3].

Existing approaches to BAX rely on expected information gain (EIG) as the criterion for selecting which points to evaluate [3]. However, maximizing the EIG presents a significant computational

---

[*]Equal contribution.

38th Conference on Neural Information Processing Systems (NeurIPS 2024).

challenge, particularly in high-dimensional problems or when the property of interest is complex. As a result, heuristic approximations are frequently employed, which can lead to suboptimal performance and limit the applicability of BAX in real-world scenarios.

To address this challenge, we propose *PS-BAX*, a simple yet effective and scalable approach based on posterior sampling. Our approach is built upon the key observation that many real-world BAX tasks aim to find a *target set*. For instance, in BO, the goal is to locate the function's global optimum, while in level set estimation, the objective is to find the points whose function value is above a specified threshold. PS-BAX only requires a single base algorithm execution at each iteration, making it much faster than EIG-based approaches, which require executing the base algorithm multiple times and optimizing over the input space. Despite its simple computation, we show that PS-BAX is competitive with existing baselines while being significantly faster. Additionally, we show that it enjoys appealing theoretical guarantees. Specifically, we prove that PS-BAX is asymptotically convergent mild regularity conditions.

Our contributions are summarized as follows:

- We derive PS-BAX, a posterior sampling-based BAX method applicable to a broad class of BAX problems, unlocking new applications and offering a fresh perspective on the scope of posterior sampling algorithms.
- We show that PS-BAX is orders of magnitude faster to compute than the EIG-based approach INFO-BAX [3] while remaining competitive with this and other specialized algorithms.
- We prove that PS-BAX is asymptotically convergent under mild regularity conditions.

## 2 Related Work

Our work falls within the broader field of probabilistic numerics [4], which frames numerical problems, such as optimization or integration, as probabilistic inference tasks. This probabilistic perspective enables uncertainty quantification, which is particularly important in settings with limited computational budgets, where budget allocation must be carefully planned, often adaptively. While much of the recent work in probabilistic numerics has focused on (Bayesian) optimization [1, 5], there have also been efforts in other areas, including integration (Bayesian quadrature) [6–8], level set estimation [2, 9], and solving differential equations [10, 11].

Recently, [3] proposed INFO-BAX, an approach to estimate an arbitrary property of interest that could be computed by a known base algorithm. Since the base algorithm requires a potentially large number of function evaluations, it cannot be applied directly. Instead, following the probabilistic numerics paradigm, a Bayesian probabilistic model of the function is used to iteratively select new points to evaluate. At each iteration, the next evaluation point is chosen by maximizing the expected information gain (EIG) between the function's value at the point and the property of interest. Similar EIG-based approaches have been employed in statistical design of experiments [12–14] and BO [15–17], often yielding excellent performance. However, these methods are computationally demanding due to the look-ahead nature of the EIG computation. Moreover, in most cases, the EIG cannot be computed in closed form and must be approximated via Monte Carlo sampling. As a result, EIG-based approaches are mainly useful in low-dimensional settings or when function evaluations are highly expensive, limiting their applicability in real-world problems.

In response to the limitations of EIG-based approaches, we explore an alternative family of strategies known as posterior sampling or Thompson sampling [18, 19]. Posterior sampling algorithms have been widely used in BO [20–22], multi-armed bandits [23–25], and reinforcement learning [26–28]. In such settings, these approaches select a point at each iteration according to the posterior probability of being the optimum. To our knowledge, our work represents the first extension of posterior sampling beyond optimization settings, offering fresh insights into this algorithmic family. While the range of problems our approach can address is narrower than those that EIG-based methods can conceptually tackle, it still encompasses a substantial class. Notably, this includes the problems explored empirically by [3] and follow-up work [29], among others.

Our work aligns with recent efforts to broaden the applicability of BO to complex real-world problems. Many such problems deviate from classical optimization formulations, exhibiting diverse structures such as combinatorial [29], robust [30, 31], or multi-level optimization [32]. Traditional BO algorithms often fail to naturally accommodate these structures, limiting their practical utility. We

**Algorithm 1** PS-BAX

---

**Require:** $p(f)$ (prior), $\mathcal{D}_0$ (initial dataset), $\mathcal{A}$ (base algorithm), $N$ (number of iterations).
 1: **for** $n = 1 : N$ **do**
 2:     Sample $\tilde{f}_n$ from $p(f \mid \mathcal{D}_{n-1})$
 3:     Apply algorithm $\mathcal{A}$ on $\tilde{f}_n$ to obtain $X_n = \mathcal{O}_{\mathcal{A}}(\tilde{f}_n)$
 4:     Choose $x_n \in \operatorname{argmax}_{x \in X_n} \mathbf{H}[f(x)|\mathcal{D}_{n-1}]$   *//Choose $x_n \in X_n$ with the highest uncertainty*
 5:     Evaluate $y_n = f(x_n) + \epsilon_n$ and set $\mathcal{D}_n = \mathcal{D}_{n-1} \cup \{(x_n, y_n)\}$
 6: **end for**
 7: **return**  Estimate of $\mathcal{O}_{\mathcal{A}}(f)$ based on $p(f \mid \mathcal{D}_N)$.      *//E.g., run $\mathcal{A}$ on the final posterior mean*

---

introduce a straightforward algorithm applicable to these diverse settings, providing a robust baseline for future exploration. Furthermore, our approach benefits from recent advances in probabilistic modeling tools [33–35], paving the way for applying these tools to a broader range of problems.

## 3   Bayesian Algorithm Execution via Posterior Sampling

**Problem Setting**   Our work takes place within the Bayesian algorithm execution (BAX) framework introduced by [3]. The goal is to estimate $\mathcal{O}_{\mathcal{A}}(f)$, the output of a *base algorithm* $\mathcal{A}$ applied to a function $f : \mathbb{X} \to \mathbb{R}$. We assume that $f$ is expensive to evaluate, which means that employing $\mathcal{A}$ directly on $f$ is infeasible (as it would require evaluating $f$ too many times). Instead, we select the points at which $f$ is evaluated sequentially, aided by a probabilistic model described below. We specifically focus on problems where the property of interest can be encoded by a set $\mathcal{O}_{\mathcal{A}}(f) \subset \mathbb{X}$, which we term the *target set*. As we shall see later, our framework encompasses a wide range of problems, including BO[2], level-set estimation, shortest-path finding on graphs, and top-$k$ estimation, with applications to topographic estimation and drug discovery.

**Probabilistic Model**   Similar to many probabilistic numerical methods, our algorithm relies on a probabilistic model encoded by a prior distribution over $f$, which we denote by $p$. Although our framework is more general and can be used with other priors, we assume for concreteness that $f$ follows a Gaussian process (GP) prior [36]. Let $\mathcal{D}_{n-1} = \{(x_k, y_k)\}_{k=1}^{n-1}$ denote the data collected after $n-1$ evaluations of $f$. We assume these evaluations are corrupted with i.i.d. Gaussian noise, i.e., $y_k = f(x_k) + \epsilon_k$, where $\epsilon_1, \ldots, \epsilon_{n-1}$ are i.i.d. with common distribution $\mathcal{N}(0, \sigma^2)$, and $\sigma^2$ is a non-negative scalar. Under these assumptions, the posterior distribution over $f$ given $\mathcal{D}_{n-1}$, denoted by $p(f \mid \mathcal{D}_{n-1})$, is again a GP whose mean and covariance functions can be computed in closed form using the classical GP regression equations.

**INFO-BAX and its Shortcomings**   Before introducing our algorithm, we briefly comment on prior work based on the expected information gain (EIG) [3]. Succinctly, the INFO-BAX approach proposed by [3] selects at each iteration the point that maximizes the expected entropy reduction between the function's value at the evaluated point and $\mathcal{O}_{\mathcal{A}}(f)$. Evaluating an expectation is generally difficult, and one often resorts to Monte Carlo sampling. Moreover, computing the EIG specifically requires expensive calculations of conditional posterior distributions and entropy. These computational issues are also present in similar information-theoretic acquisition functions proposed in the classic BO setting. However, for BAX tasks, the computation burden of EIG can be much more pronounced if $|\mathcal{O}_{\mathcal{A}}(f)|$ is large. This occurs, for example, in the level set estimation setting, where $\mathcal{O}_{\mathcal{A}}(f)$ can be comprised of a large number of points. We defer a more detailed discussion of the computation of the EIG to Appendices A and B .

**PS-BAX**   To overcome the computational limitations of EIG-based approaches, we introduce a simple strategy based on posterior sampling, which we term PS-BAX. For ease of exposition, we only describe the fully sequential version of our algorithm and defer the batched (parallelized) version to Appendix C. Our algorithm is summarized in Algorithm 1. PS-BAX is comprised of two steps, detailed below.

---

[2]To reduce to standard BO, one can define the target set $\mathcal{O}_{\mathcal{A}}(f)$ as the points $x \in \mathbb{X}$ that maximize $f$, i.e., $\mathcal{O}_{\mathcal{A}}(f) = \operatorname{argmax}_{x \in \mathbb{X}} f(x)$ (often a singleton set).

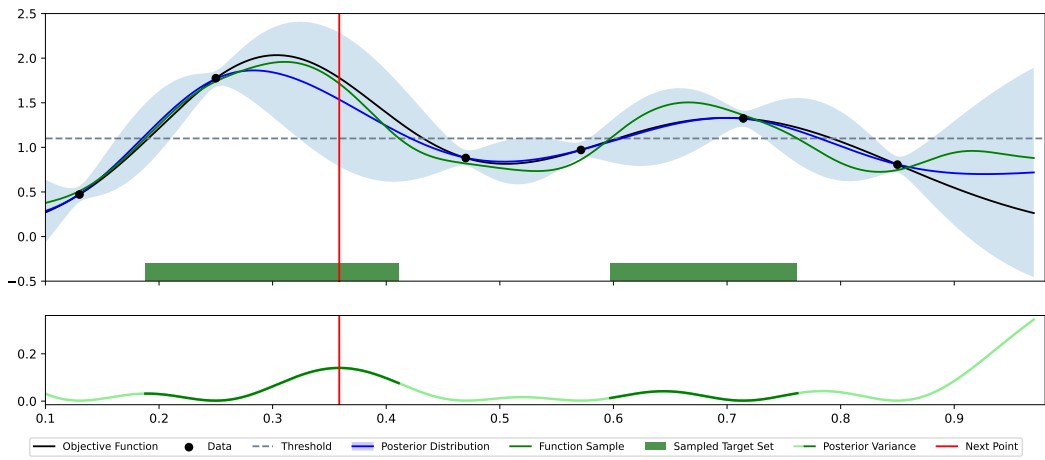

Figure 1: Depiction of PS-BAX (Algorithm 1) for the level-set estimation problem. We plot the objective function $f$ (black line), the current available data $\mathcal{D}_{n-1}$ (black points), the threshold (grey dashed line), the posterior distribution $p(f \mid \mathcal{D}_{n-1})$ (blue line and light blue region), a sample from the posterior $\tilde{f}_n \sim p(f \mid \mathcal{D}_{n-1})$ (green line), the corresponding sampled target set $X_n = \mathcal{O}_{\mathcal{A}}(\tilde{f}_n)$ (green region) (this is the set of inputs where the green line is above the threshold), the variance of $p(f \mid \mathcal{D}_{n-1})$ (green line, bottom row), and the next point to evaluate selected by PS-BAX $x_n \in X_n$ (input marked by the vertical red line). The key step is computing the target set $X_n$ using the sampled function $\tilde{f}_n$, which generalizes posterior sampling for standard BO.

1. We sample a target set $X_n \subset \mathbb{X}$ according to the posterior probability that $X_n = \mathcal{O}_{\mathcal{A}}(f)$ (lines 2-3 in Algorithm 1). This can be achieved by drawing a sample from the posterior over $f$, denoted by $\tilde{f}_n$ (line 2), and then setting $X_n = \mathcal{O}_{\mathcal{A}}(\tilde{f}_n)$ (line 3).
2. We select the point in the sampled target set $X_n$ with maximal uncertainty or entropy: $x_n \in \operatorname{argmax}_{x \in X_n} \mathbf{H}[f(x)|\mathcal{D}_n]$ (line 4 in Algorithm 1). For a Gaussian posterior, $x_n$ can be equivalently selected using the maximal posterior standard deviation: $x_n \in \operatorname{argmax}_{x \in X_n} \sigma_n(x)$, where $\sigma_n(x)$ is the posterior standard deviation of $f(x)$.

Note that the second step is unnecessary in standard BO, since $X_n$ is typically a singleton.

**Depiction for Level Set Estimation**  Figure 1 depicts an iteration of PS-BAX for the level-set estimation problem, where $\mathcal{O}_{\mathcal{A}}(f) := \{x \in \mathbb{X} \mid f(x) > \tau\}$, for a user-specified value of $\tau$. Line 3 of Algorithm 1 returns a target set $X_n$ based on where the sampled function $\tilde{f}_n$ (green line in Figure 1) is above the threshold $\tau$ (green region in Figure 1). Line 4 of Algorithm 1 then chooses the point $x_n \in X_n$ that has maximal uncertainty (red line in Figure 1). In the standard BO setting where $\mathcal{A}$ is computing the maximizer of $f$, the target region $X_n$ is simply a singleton point where the sampled $\tilde{f}$ has highest value (and Line 4 in Algorithm 1 is not necessary).

**Discussion**  We now provide an intuitive explanation of why one might expect PS-BAX to perform well. In the standard BO setting, posterior sampling is known to deliver excellent performance [20, 37] and enjoys strong theoretical guarantees [18, 20]. Like in the BO setting, the intrinsic goal of posterior sampling in our setting is to balance exploration and exploitation. In our case, this means selecting points for which, according to our probabilistic model, membership in the target set is still highly uncertain among the likely candidates. To achieve this, the first step of PS-BAX selects a random set $X_n$, according to the probability of this set being the target set, in the same vein as traditional posterior sampling in the BO setting (the set of likely candidates). Unlike in BO, however, the target set is, in principle, comprised of several points, and thus, we must come up with a criterion to choose one. To overcome this, the second step simply selects the point with the highest *uncertainty* among points in $X_n$, which is a standard strategy in the active learning literature [38].

**Computational Efficiency**  PS-BAX requires running $\mathcal{A}$ only once on a single sample of $f$, contributing to its practicality and scalability. Furthermore, similar to posterior sampling in the standard

BO setting, PS-BAX avoids the need to maximize an acquisition function over $\mathbb{X}$, a process that is computationally expensive because it involves calculating the expected value of quantities like information gain. As demonstrated in our experiments, this makes PS-BAX significantly faster than INFO-BAX [3], particularly in problems where either $\mathcal{O}_\mathcal{A}(f)$ or $\mathbb{X}$ are large.

**Convergence of PS-BAX** A natural question is under which conditions is PS-BAX able to *find* the target set given enough evaluations. We address this question below. Before stating our results, we introduce a definition related to the characterization of problems where PS-BAX converges.

**Definition 1.** *A target set estimated by an algorithm $\mathcal{A}$ is said to be complement-independent if, for any pair of functions $f, f' : \mathbb{X} \to \mathbb{R}$, it holds that $\mathcal{O}_\mathcal{A}(f) = \mathcal{O}_\mathcal{A}(f')$ whenever there exists a set $S$ such that $\mathcal{O}_\mathcal{A}(f) \cup \mathcal{O}_\mathcal{A}(f') \subset S$ and $f(x) = f'(x)$ for all $x \in S$.*

Many target sets of interest, such as a function's optimum or level set, are complement-independent. Indeed, the value of $f$ at points that are not the optimum or that do not lie in the level of interest do not influence these properties. Theorem 1 below shows that PS-BAX enjoys Bayesian posterior concentration, provided the target set of interest is complement-independent. Intuitively, this result means that if $f$ is drawn from the prior used by our algorithm (i.e., the prior is well-specified), then, with probability one, the posterior will concentrate around the true target set. Corollary 1 gives an asymptotically consistent estimator of the target set. Finally, we also show there are problems where the target set is not complement-independent and PS-BAX is not asymptotically consistent in Theorem 2. The proofs of these results can be found in Appendix D.

**Theorem 1.** *Suppose that $\mathbb{X}$ is finite and that the target set estimated by $\mathcal{A}$ is complement-independent. If the sequence of points $\{x_n\}_{n=1}^\infty$ is chosen according to the PS-BAX algorithm, then, for each $X \subset \mathbb{X}$, $\lim_{n \to \infty} \mathbf{P}_n(\mathcal{O}_\mathcal{A}(f) = X) = \mathbf{1}\{\mathcal{O}_\mathcal{A}(f) = X\}$ almost surely for $f$ drawn from the prior.*

**Corollary 1.** *Suppose the assumptions of Theorem 1 hold, and let $T_n \in \mathrm{argmax}_{X \subset \mathbb{X}} \mathbf{P}_n(\mathcal{O}_\mathcal{A}(f) = X)$. Then, for $f$ drawn from the prior, we have $T_n = \mathcal{O}_\mathcal{A}(f)$ for all sufficiently large $n$ almost surely.*

**Theorem 2.** *There exists a problem instance (i.e., $\mathbb{X}$, a Bayesian prior over $f$, and $\mathcal{A}$) such that if the sequence of points $\{x_n\}_{n=1}^\infty$ is chosen according to the PS-BAX algorithm, then there is a set $X \subset \mathbb{X}$ such that $\lim_{n \to \infty} \mathbf{P}_n(\mathcal{O}_\mathcal{A}(f) = X) = 1/2$ almost surely for $f$ drawn from the prior.*

## 4 Numerical Experiments

We evaluate the performance of PS-BAX on eight problems across four problem classes. For each problem class, we specify the base algorithm used. We compare the performance of PS-BAX against INFO-BAX [3] and uniform random sampling over $\mathbb{X}$ (Random). When available, we also include an algorithm from the literature specifically designed for the problem class. Additional implementation details of the algorithms are described in Appendix E. In all experiments, an initial dataset is generated by sampling $2(d + 1)$ inputs uniformly at random from $\mathbb{X}$, where $d$ denotes the dimensionality of $\mathbb{X}$. Following this initialization, each algorithm sequentially selects additional batches of points. Unless stated otherwise, the batch size is set to $q = 1$. The performance of each algorithm is determined by applying $\mathcal{A}$ on $\mu_n$, the posterior mean of $f$ given $\mathcal{D}_n$ and subsequently computing a suitable performance metric on $\mathcal{O}_\mathcal{A}(\mu_n)$. Each experiment was replicated 30 times, with plots showing mean performance plus and minus 1.96 standard errors. Code to reproduce our experiments is available at https://github.com/RaulAstudillo06/PSBAX.

**Summary of Findings** Overall, we find that PS-BAX is always competitive with and sometimes significantly outperforms INFO-BAX across all of our experiments. Additionally, as shown in Table 1, PS-BAX can be orders of magnitude faster in wall-clock runtime. PS-BAX outperforms INFO-BAX on five out of eight problems, offering particularly large improvements in the Local Optimization and DiscoBAX problem classes. Moreover, in the Local Optimization and Level Set Estimation problems, PS-BAX also outperforms algorithms from the literature specifically designed for such problem classes. On the simpler problems, such as those in the Top-$k$ problem class, PS-BAX is competitive with INFO-BAX while still being significantly faster.

### 4.1 Local Optimization

We explore the performance of our algorithm in the local optimization setting, where $\mathcal{A}$ is a classic optimization algorithm, i.e., an algorithm designed for optimization problems where $f$ (and potentially

| Problem | PS-BAX Runtime (s) | INFO-BAX Runtime (s) |
|---|---|---|
| Local Optimization: Hartmann (6D) | 0.37 | 7.64 |
| Local Optimization: Ackley (10D) | 3.36 | 29.31 |
| Level Set Estimation: Himmelblau | 0.57 | 14.97 |
| Level Set Estimation: Volcano | 0.49 | 289.91 |
| Top-$k$: Rosenbrock ($k = 6$) | 0.92 | 18.31 |
| Top-$k$: GB1 ($k = 10$) | 145.23 | 865.85 |
| DiscoBAX: Tau Protein Assay | 3.78 | 113.20 |
| DiscoBAX: Interferon-Gamma Assay | 3.95 | 97.03 |

Table 1: Average runtimes per iteration of PS-BAX and INFO-BAX across our test problems. In all of them, PS-BAX is between one and three orders of magnitude faster than INFO-BAX. We also note that the runtimes for both algorithms are significantly longer on the Top-10 GB1 problem due to the use of a deep kernel GP model.

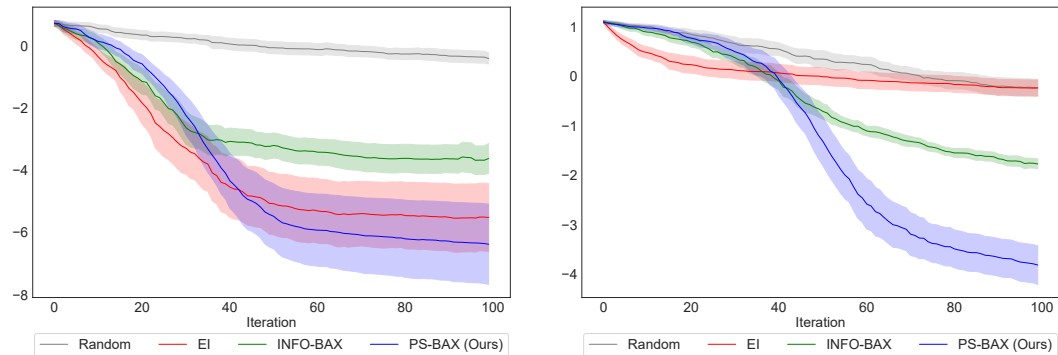

Figure 2: Results for Local Optimization, showing the log10 inference regret achieved by the compared algorithms (lower values indicate better performance). The left and right panels present results for the Hartmann-6D and Ackley-10D functions, respectively. On Hartmann-6D, PS-BAX and EI perform comparably, both outperforming INFO-BAX. On Ackley-10D, PS-BAX achieves significantly better results than the rest of the algorithms.

its gradients) can be evaluated at a large number of points. Examples of such algorithms include evolutionary algorithms [39], trust-region methods [40], and many gradient-based optimization algorithms [41, 42]. This setting reduces to the classical BO setting if $\mathcal{A}$ can recover the global optimum of $f$. In such case, the INFO-BAX reduces to the classical predictive entropy search acquisition function [16] when computed exactly and to the joint entropy search acquisition function [43] under the approximation proposed by [3] that we use in our experiments. PS-BAX, in turn, reduces to the classical posterior sampling strategy used in BO [20]. However, due to its practical relevance and the lack of an empirical comparison between joint entropy search and posterior sampling, we still include this setting in our experiments. We also use this setting to illustrate nuances that arise when choosing a base algorithm.

In our experiments, we use a gradient-based optimization method as a base algorithm instead of an evolutionary algorithm as pursued by [3]. Gradient-based methods typically exhibit faster convergence than their gradient-free counterparts. However, they are often infeasible if gradients cannot be obtained analytically and instead are obtained, e.g., via finite differences. Since in most applications, analytic gradients of $f$ are unavailable, directly applying such methods on $f$ is infeasible. However, PS-BAX and INFO-BAX can make use of gradient-based methods thanks to the availability of gradients of most probabilistic models used in practice, including GPs.

We consider the Hartmann and Ackley functions, with input dimensions of 6 and 10, respectively, as test functions. Both functions have many local minima and are standard test functions in the BO literature. For Ackley, we set the batch size to $q = 2$. As a performance metric, we report the log10 inference regret, given by $\log_{10}(f^* - f(\hat{x}_n^*))$, where $\hat{x}_n^*$ is obtained by applying $\mathcal{A}$ on $\mu_n$. The results of these experiments are depicted in Figure 2. As a baseline, we also include the

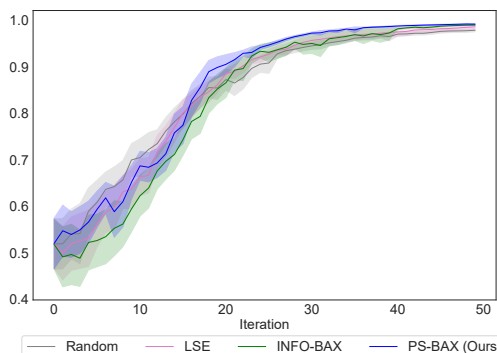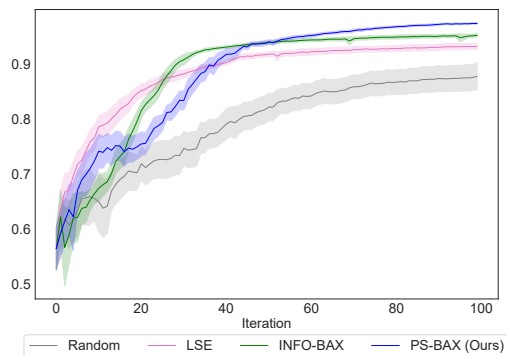

Figure 3: Results for Level Set Estimation, showing the F1 score (where higher is better). The left and right panels present results for the Himmelblau test function and the topographic mapping problem, respectively. In the former problem, all algorithms perform similarly, while in the latter, PS-BAX outperforms all baselines.

expected improvement (EI), arguably the most popular BO acquisition function. On Hartmann-6D, PS-BAX performs on pair with EI, and both algorithms outperform INFO-BAX. Notably, PS-BAX outperforms both INFO-BAX and EI on Ackley-10D.

## 4.2 Level Set Estimation

Level set estimation involves finding all points in $\mathbb{X}$ with $f(x) > \tau$, for a user-specified threshold value $\tau$. This task arises in applications such as environmental monitoring, where a mobile sensing device detects regions with dangerous pollution levels [2], and topographic mapping, where the goal is to infer the portion of a geographic area above a specified altitude using limited measurements [44]. For both problems considered in our work, $\mathbb{X}$ is finite; therefore, the base algorithm $\mathcal{A}$ simply ranks all objective values and returns the points where the function value exceeds the threshold.

We evaluate the algorithms on a synthetic problem (the 2-dimensional Himmelblau function) and a real-world topographic dataset, consisting of $87 \times 61$ height measurements from a large geographic area around Auckland's Maunga Whau volcano [44]. The threshold $\tau$ is set to the 0.55 quantile of all function values in the domain for both problems. An illustration of single runs on the topographic problem over 100 iterations for both INFO-BAX and PS-BAX is shown in Figure 4.

The performance metric used is the F1 score, defined by

$$F1 = \frac{2TP}{2TP + FP + FN},\qquad(1)$$

where $TP$, $FP$, and $FN$ represent true positives, false positives, and false negatives, respectively. The results of this experiment are shown in Figure 3. As an additional baseline specifically designed for level set estimation, we include the popular LSE algorithm proposed by [2]. PS-BAX demonstrates strong performance, outperforming all benchmarks in the topographic mapping problem.

## 4.3 Top-$k$ Estimation

We consider the top-$k$ estimation setting, where $\mathbb{X}$ is a finite (but potentially large) set, and the goal is to identify the $k$ points with the largest values of $f(x)$. In this scenario, the base algorithm evaluates $f$ at all points in $\mathbb{X}$ and returns the $k$ best points. Following [3], we use as performance metric the Jaccard distance between the estimated output $S_n = \mathcal{O}_{\mathcal{A}}(\mu_n)$ and the ground truth optimal set $S^*$, defined by

$$d(S_n, S^*) = 1 - \frac{|S_n \cap S^*|}{|S_n \cup S^*|}.\qquad(2)$$

We consider two test problems. The first problem uses 3-dimensional Rosenbrock function, a standard benchmark in the optimization literature. The input space is obtained by taking a uniform grid of 1,000 points over $[-2, 2]^3$. For this problem we set $k = 4$.

The second problem is a real-world top-$k$ ($k = 10$) selection task in protein design, where the goal is to maximize stability fitness predictions for the Guanine nucleotide-binding protein GB1,

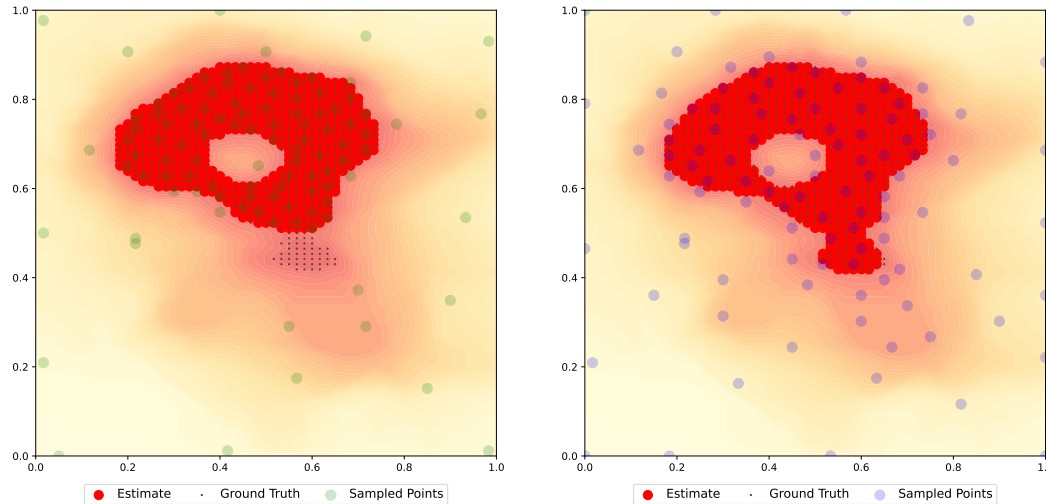

Figure 4: Depiction of the INFO-BAX (left) and PS-BAX (right) algorithms on the topographic level set estimation problem described in Section 4.2. Each figure shows the ground truth super-level set (small black dots), the points evaluated after 100 iterations (green and blue dots for INFO-BAX and PS-BAX, respectively), and the estimated level set from the final posterior mean (red dots). PS-BAX provides an accurate estimate of the level set, whereas INFO-BAX misses a significant portion.

given different sequence mutations in a target region of 4 residues [45]. GB1 is well-studied by biologists, and its domain is known to be highly rugged, dominated by "dead" variants with very low fitness scores [46]. There are $20^4$ possible combinations, with 20 amino acids and 4 positions, and we represent the input space $\mathbb{X}$ as one-hot vectors in an 80-dimensional space. To avoid excessive runtimes, we randomly sample 10,000 points from the original dataset. Due to the high dimensionality, vast input space, and sparse fitness landscape, this dataset poses significant challenges for standard GP models. Therefore, we use a deep kernel GP [47] as our probabilistic model. Given the dataset's size, we perform batched evaluations with batch size of $q = 4$ for both PS-BAX and INFO-BAX.

The results of these experiments are shown in Figure 5. In both problems, PS-BAX performs comparably to INFO-BAX, with both algorithms significantly outperforming Random.

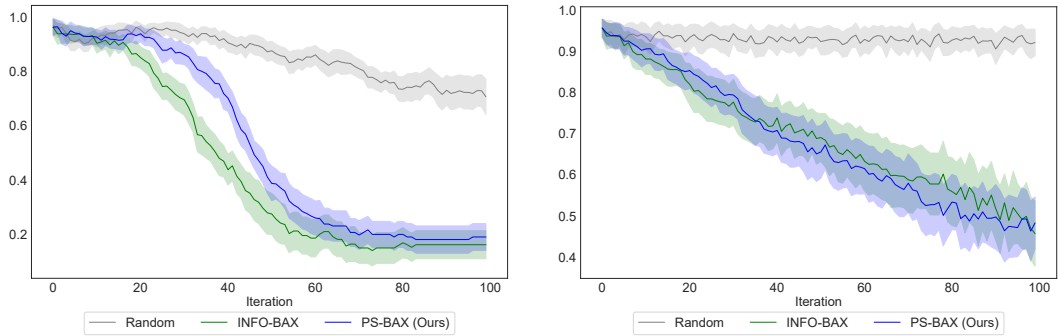

Figure 5: Results for Top-$k$ Estimation, showing the Jaccard distance (where lower is better). The left panel presents results for the 3-dimensional Rosenbrock test function with $k = 4$, while the right panel shows results for the real-world protein design GB1 dataset with $k = 10$. In both problems, PS-BAX performs similarly to INFO-BAX.

## 4.4 DiscoBAX: Drug Discovery Application

As a final application, we consider the DiscoBAX problem setting from [29] in the context of drug discovery, where the task is to identify a set of optimal genomic interventions to determine suitable drug targets. Formally, let $\mathbb{X}$ represent a pool of genetic interventions, and for each $x \in \mathbb{X}$, let $f(x)$

denote an in vitro phenotype measurement correlated with the effectiveness of genetic intervention $x$. The effectiveness of the intervention is assumed to be $f(x) + \eta(x)$, where $\eta(x)$ captures noise and other exogenous factors not reflected in the in vitro measurement. Following the setup in [29], we simulate $\eta$ using a GP with mean 0 and an RBF covariance function. The goal is to identify a small set of genomic interventions in $\mathbb{X}$ that maximize an objective function balancing two characteristics: high expected change in the target phenotype and high diversity to maximize success in subsequent stages of drug development. This is formalized in [29] as the following optimization problem:

$$\max_{S \subset \mathbb{X}:|S|=k} \mathbb{E}_\eta \left[ \max_{x \in S} f(x) + \eta(x) \right],\tag{3}$$

where $k$ is the desired size of the intervention set. This problem aims to find a set of interventions $S$ such that the best-performing intervention in $S$ has the highest expected effectiveness (over $\eta$). Solving Equation 3 exactly is challenging due to its combinatorial nature, even if could evaluate $f$ many times, but a computationally efficient approximation is possible by leveraging the submodularity of the objective function. For more details on the base algorithm, we refer the reader to [29].

Following [29], we use the tau protein assay [48] and interferon-gamma assay [49] datasets from the Achilles project [50]. Originally, the gene embeddings in this dataset are represented as 808-dimensional vectors, and a Bayesian MLP is used as the probabilistic model instead of a GP. To reduce dimensionality, we preprocess the dataset using Principal Component Analysis (PCA) and then fit a GP to the lower-dimensional representation. Additionally, we truncate the dataset to the 5000 genes with the highest intervention values to ensure computational feasibility in our experiments. The results of these experiments are shown in Figure 6. PS-BAX significantly outperforms INFO-BAX, whose performance is only marginally better than that of Random.

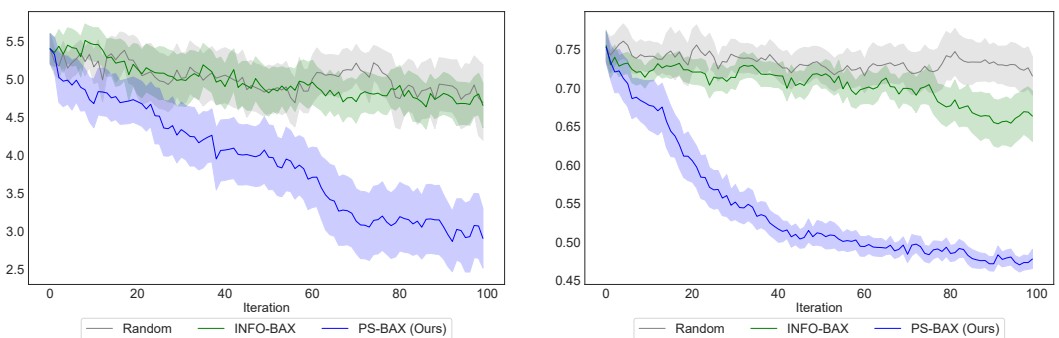

Figure 6: Results for DiscoBAX [29], showing the regret between the solution found by applying a greedy submodular optimization algorithm to the objective in Equation 3 and the solution obtained from applying the same algorithm over the posterior mean instead of the true function. Both problems are based on the Achilles dataset [50], with the left panel presenting results for the tau protein assay [48] and the right panel showing results for the interferon-gamma assay. In both cases, PS-BAX significantly outperforms INFO-BAX, which performs only marginally better than Random.

## 5   Conclusion

Many real-world problems involve estimating the output of a base algorithm applied to a black-box function with costly evaluations. While the INFO-BAX algorithm proposed by [3] offers a solution, it faces practical limitations. In response, we introduced PS-BAX, a novel posterior sampling strategy built upon the observation that, in many cases, the algorithm's output can be characterized as a target set of input points. Our experiments demonstrate that PS-BAX is not only competitive with previous approaches but also significantly faster to compute. Moreover, we established conditions under which PS-BAX is asymptotically convergent.

Looking ahead, our approach provides a pathway to extend the success of Bayesian optimization to a broader range of problems, potentially unlocking new and impactful applications. Additionally, PS-BAX serves as a robust baseline for future research aimed at developing tailored strategies for specific domains. Furthermore, our findings offer new perspectives on posterior sampling algorithms and their application scope, suggesting several promising avenues for future exploration in this area.

## Acknowledgments and Disclosure of Funding

This work was performed under the auspices of the U.S. Department of Energy by Lawrence Livermore National Laboratory under Contract DE-AC52-07NA27344. LLNL-CONF-864204. The GUIDE program is executed by the Joint Program Executive Office for Chemical, Biological, Radiological and Nuclear Defense's (JPEO-CBRND) Joint Project Lead for CBRND Enabling Biotechnologies (JPL CBRND EB) on behalf of the Department of Defense's Chemical and Biological Defense Program. This effort was in collaboration with the Defense Health Agency (DHA) COVID funding initiative. The views expressed in this paper reflect the views of the authors and do not necessarily reflect the position of the Department of the Army, Department of Defense, nor the United States Government. References to non-federal entities do not constitute nor imply Department of Defense or Army endorsement of any company or organization.

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

## A   Computation of the Expected Information Gain

Let $\mathbf{E}$ and $\mathbf{H}$ denote the expectation and (differential) entropy operators, respectively. At each iteration $n$, the expected information gain between the $\mathcal{O}_{\mathcal{A}}(f)$ and a new observation of $f$ at $x$, denoted by $y_x$, can be written as

$$\mathrm{EIG}_n(x) = \mathbf{H}[y_x \mid \mathcal{D}_n] - \mathbf{E}[\mathbf{H}[y_x \mid \mathcal{D}_n, \mathcal{O}_{\mathcal{A}}(f)] \mid \mathcal{D}_n]. \tag{4}$$

Under the probabilistic model established above, the conditional distribution of $y_x$ given $\mathcal{D}_n$ is Gaussian, allowing the analytical computation of $\mathbf{H}[y_x \mid \mathcal{D}_n]$. However, in most cases, $\mathbf{H}[y_x \mid \mathcal{D}_n, \mathcal{O}_{\mathcal{A}}(f)]$ cannot be computed analytically. In particular, this is true in our setting, where $\mathcal{O}_{\mathcal{A}}(f)$ is a subset of $\mathbb{X}$.

To address this challenge, [3] introduced an approximation where $\mathcal{O}_{\mathcal{A}}(f)$ is replaced by a set of pairs $(x', f(x'))$ for inputs $x'$ evaluated when $\mathcal{A}$ is applied on $f$. When $\mathcal{O}_{\mathcal{A}}(f)$ is a subset of $\mathbb{X}$, a natural choice is to take all inputs $x' \in \mathcal{O}_{\mathcal{A}}(f)$. This coincides with the approximation used by [3] in the optimization and top-$k$ estimation settings. The corresponding approximation of $\mathrm{EIG}_n$, denoted by $\mathrm{EIG}_n^v$, is then given by

$$\mathrm{EIG}_n^v(x) = \mathbf{H}[y_x \mid \mathcal{D}_n] - \mathbf{E}[\mathbf{H}[y_x \mid \mathcal{D}_n, \{(x', f(x')) : x' \in \mathcal{O}_{\mathcal{A}}(f)\}] \mid \mathcal{D}_n]. \tag{5}$$

The advantage of this approximation is that, again, $\mathbf{H}[y_x \mid \mathcal{D}_n, \{(x', f(x')) : x' \in \mathcal{O}_{\mathcal{A}}(f))\}$ can be computed analytically under a Gaussian posterior.

The expectation $\mathbf{E}[\mathbf{H}[y_x \mid \mathcal{D}_n, \{(x', f(x')) : x' \in \mathcal{O}_{\mathcal{A}}(f)\}] \mid \mathcal{D}_n]$ still requires to be approximated via Monte Carlo sampling. Concretely, this can be achieved by drawing $L$ samples from the posterior over $f$ given $\mathcal{D}_n$, denoted by $\tilde{f}_{n,1}, \ldots, \tilde{f}_{n,L}$, and setting

$$\mathrm{EIG}_n^v(x) \approx \mathbf{H}[y_x \mid \mathcal{D}_n] - \frac{1}{L}\sum_{\ell=1}^{L} \mathbf{H}[y_x \mid \mathcal{D}_n, \{(x', f(x')) : x' \in \mathcal{O}_{\mathcal{A}}(\tilde{f}_{n,\ell})\}]. \tag{6}$$

This is the approximation of $\mathrm{EIG}_n$ that we use in our experiments in Section 4, i.e., at each iteration, we set $x_n$ to be a point that maximizes the expression in Equation 6.

# B Computational Complexity of PS-BAX and INFO-BAX

Given a Gaussian process posterior, we analyze the computational complexity of selecting the next evaluation point for both PS-BAX and INFO-BAX. Our analysis excludes the cost of generating a sample from the posterior, which is fixed and depends only on the number of Fourier features used. We also assume that the cost of evaluating such a sample at any given point is 1, as is the cost of evaluating the posterior mean and covariance. Additionally, we assume that the function domain $\mathbb{X}$ is discrete with $|\mathbb{X}| = N$, the algorithm output has a fixed cardinality $|\mathcal{O}_\mathcal{A}(f)| = M$, the number of execution paths to approximate the posterior entropy is $L$, and running the algorithm on any input function requires $P$ evaluations. As we show next, the computational cost of INFO-BAX can be significantly higher than that of PS-BAX when $N$, $M$, or $L$ is large.

For PS-BAX, the complexity is $O(P + M)$, reflecting the cost of running the algorithm once on a single function sample and maximizing the posterior variance over the sampled target set. For INFO-BAX, the complexity is $O((P + M^3 + N \cdot M^2) \cdot L)$. For each function sample, we need to execute the algorithm ($P$), condition on $M$ new points to compute the conditional entropy ($M^3$), and evaluate the posterior variance of the fantasized model on $N$ points ($N \cdot M^2$). This process is repeated for $L$ function samples.

# C Batch Extensions of PS-BAX and INFO-BAX

In this section, we discuss extensions of the PS-BAX and INFO-BAX algorithms to the batch setting, where at each iteration, we generate $q$ new points for evaluation, denoted by $x_{n,1}, \ldots, x_{n,q}$. These extensions are inspired by batch versions of the posterior sampling algorithm [20] and the joint entropy search acquisition function [43] from BO. Figure 7 illustrates the performance of PS-BAX under various batch sizes in two of our test problems.

**Batch PS-BAX**   We extend PS-BAX to the batch setting, following the approach proposed by [20]. For a batch size of $q$, we draw $q$ independent samples from the posterior on $f$, denoted by $\tilde{f}_1, \ldots, \tilde{f}_q$, and define the set $X_n = \cup_{i=1}^{q} \mathcal{O}_\mathcal{A}(\tilde{f}_{n,i})$. We then select the points $x_{n,1}, \ldots, x_{n,q} \in X_n$ iteratively by choosing the point in $X_n$ with the highest posterior entropy, conditioned on the previously selected points, as follows:

$$x_{n,1} = \mathrm{argmax}_{x \in X_n} \mathbf{H}[f(x) \mid \mathcal{D}_n],$$
$$\vdots$$
$$x_{n,q} = \mathrm{argmax}_{x \in X_n} \mathbf{H}[f(x) \mid \mathcal{D}_n \cup \{x_{n,1}, \ldots, x_{n,q-1}\}]$$

**Batch INFO-BAX**   INFO-BAX can be naturally extended to the batch setting by considering the EIG over a batch of $q$ points. However, directly optimizing the EIG in this scenario involves optimizing over $\mathbb{X}^q$, which is usually computationally prohibitive. To address this, we use a greedy optimization approach. Specifically, we select $x_{n,1}, \ldots, x_{n,q}$ iteratively by maximizing the EIG conditioned on the previously selected points, as follows:

$$x_{n,1} = \mathrm{argmax}_{x \in \mathbb{X}} \mathbf{H}[y_x \mid \mathcal{D}_n] - \mathbf{E}[\mathbf{H}[y_x \mid \mathcal{D}_n, \mathcal{O}_\mathcal{A}(f)] \mid \mathcal{D}_n],$$
$$\vdots$$
$$x_{n,q} = \mathrm{argmax}_{x \in \mathbb{X}} \mathbf{H}[y_x \mid \mathcal{D}_n \cup \{x_{n,1}, \ldots, x_{n,q-1}\}]$$
$$- \mathbf{E}[\mathbf{H}[y_x \mid \mathcal{D}_n \cup \{x_{n,1}, \ldots, x_{n,q-1}\}, \mathcal{O}_\mathcal{A}(f)] \mid \mathcal{D}_n \cup \{x_{n,1}, \ldots, x_{n,q-1}\}].$$

# D Proofs of Theorems 1 and 2

We begin by introducing the following notation. Let $\mathcal{F}_n$ denote the $\sigma$-algebra generated by $\mathcal{D}_{n-1}$, and let $\mathcal{F}_\infty$ denote the minimal $\sigma$-algebra generated by the sequence $\{\mathcal{F}_n\}_{n=1}^{\infty}$. We denote the conditional probability measures induced by $\mathcal{F}_n$ and $\mathcal{F}_\infty$ as $\mathbf{P}_n$ and $\mathbf{P}_\infty$, respectively.

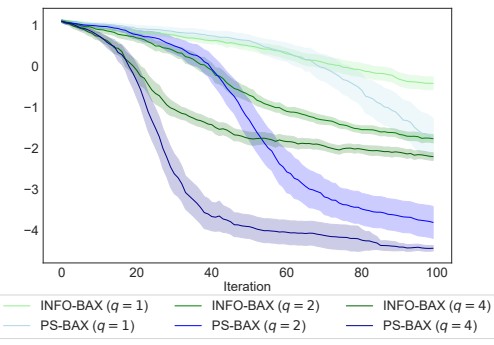 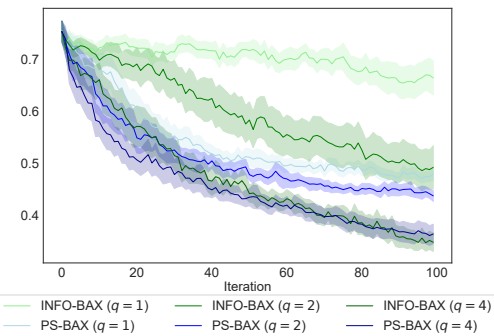

Figure 7: Performance of PS-BAX and INFO-BAX under batch sizes $q = 1, 2, 4$ on the local optimization Ackley-10D problem (left) and the DiscoBAX interferon-gamma essay problem (right).

### D.1 Proof of Theorem 1

Before proving Theorem 1, we establish an auxiliary lemma. Both Lemma 1 and Theorem 1 assume the prior distribution and likelihood discussed in Section 3. In particular, this implies that for each $x \in \mathbb{X}$, the posterior distribution of $f(x)$ given $\mathcal{F}_n$ is Gaussian, with mean and standard deviation denoted by $\mu_n(x)$ and $\sigma_n(x)$, respectively.

**Lemma 1.** *Suppose that $\mathbb{X}$ is finite, and let $S_\infty = \{x \in \mathbb{X} : \exists\, X \subset \mathbb{X} \text{ s.t. } x \in X \wedge \mathbf{P}_\infty(\mathcal{O}_\mathcal{A}(f) = X) > 0\}$. Then, both $S_\infty$ and the image of $f$ over this set are $\mathcal{F}_\infty$-measurable.*

*Proof.* Clearly, the set $S_\infty$ is $\mathcal{F}_\infty$-measurable. Thus, it suffices to show that for each $x \in S_\infty$, $f(x)$ is $\mathcal{F}_\infty$-measurable.

Let $x \in S_\infty$. By definition, there exists a subset $X \subset \mathbb{X}$ such that $\mathbf{P}_\infty(\mathcal{O}_\mathcal{A}(f) = X) > 0$. Moreover, a standard martingale argument shows that $\lim_{n\to\infty} \mathbf{P}_n(\mathcal{O}_\mathcal{A}(f) = X) = \mathbf{P}_\infty(\mathcal{O}_\mathcal{A}(f) = X)$ almost surely. Consequently, there exists $\epsilon > 0$ such that $\mathbf{P}_n(\mathcal{O}_\mathcal{A}(f) = X) > \epsilon$ for all sufficiently large $n$, implying that the event $X_n = X$ occurs infinitely often.

It is not hard to see that $\sigma_n(x) \to 0$ as $n \to \infty$ if $x$ is selected infinitely often. Furthermore, since $x_n = \operatorname{argmax}_{x' \in X_n} \sigma_n(x')$, $\mathbb{X}$ is finite, and $X_n = X$ occurs infinitely often, it follows that $\sigma_n(x') \to 0$ as $n \to \infty$ for each $x' \in X$, and in particular for $x$. By Proposition 2.9 in [51], it follows that $\mu_n(x) \to f(x)$. Since $\mu_n(x)$ is $\mathcal{F}_\infty$-measurable for each $n$, it follows that $f(x)$ is $\mathcal{F}_\infty$-measurable for each $x \in S_\infty$. $\qquad\square$

**Theorem 1.** *Suppose that $\mathbb{X}$ is finite and that the target set estimated by $\mathcal{A}$ is complement-independent. If the sequence of points $\{x_n\}_{n=1}^\infty$ is chosen according to the PS-BAX algorithm, then for each $X \subset \mathbb{X}$, $\lim_{n\to\infty} \mathbf{P}_n(\mathcal{O}_\mathcal{A}(f) = X) = \mathbf{1}\{\mathcal{O}_\mathcal{A}(f) = X\}$ almost surely for $f$ drawn from the prior.*

*Proof.* Recall that $\lim_{n\to\infty} \mathbf{P}_n(\mathcal{O}_\mathcal{A}(f) = X) = \mathbf{P}_\infty(\mathcal{O}_\mathcal{A}(f) = X)$ almost surely. Thus, it remains to show that $\mathbf{P}_\infty(\mathcal{O}_\mathcal{A}(f) = X) = \mathbf{1}\{\mathcal{O}_\mathcal{A}(f) = X\}$ almost surely.

Let $S_\infty$ be defined as in Lemma 1. Since $\mathbb{X}$ is finite, by the definition of $S_\infty$, it follows that $\mathbf{P}_\infty(\mathcal{O}_\mathcal{A}(f) \subset S_\infty) = 1$. Moreover, by the law of iterated expectation, it also holds that $\mathbf{P}(\mathcal{O}_\mathcal{A}(f) \subset S_\infty) = 1$.

Since $\mathcal{O}_\mathcal{A}(f)$ is complement-independent, it is fully determined by the values of $f$ over $S_\infty$ whenever $\mathcal{O}_\mathcal{A}(f) \subset S_\infty$. By Lemma 1, we know that both $S_\infty$ and the image of $f$ over this set are $\mathcal{F}_\infty$-measurable. Therefore, for any fixed $X \subset \mathbb{X}$, we have $\{\mathcal{O}_\mathcal{A}(f) = X\} \cap \{\mathcal{O}_\mathcal{A}(f) \subset S_\infty\} \in \mathcal{F}_\infty$, and hence, $\mathbf{P}_\infty(\mathcal{O}_\mathcal{A}(f) = X, \mathcal{O}_\mathcal{A}(f) \subset S_\infty) = \mathbf{1}\{\mathcal{O}_\mathcal{A}(f) = X\}\mathbf{1}\{\mathcal{O}_\mathcal{A}(f) \subset S_\infty\}$.

Finally, since $\mathbf{P}(\mathcal{O}_\mathcal{A}(f) \subset S_\infty) = \mathbf{P}_\infty(\mathcal{O}_\mathcal{A}(f) \subset S_\infty) = 1$, we conclude that $\mathbf{P}_\infty(\mathcal{O}_\mathcal{A}(f) = X) = \mathbf{1}\{\mathcal{O}_\mathcal{A}(f) = X\}$ almost surely. $\qquad\square$

### D.2 Proof of Theorem 2

**Theorem 2.** *There exists a problem instance (i.e., $\mathbb{X}$, a Bayesian prior over $f$, and $\mathcal{A}$) such that if the sequence of points $\{x_n\}_{n=1}^{\infty}$ is chosen according to the PS-BAX algorithm, then there is a set $X \subset \mathbb{X}$ such that $\lim_{n\to\infty} \mathbf{P}_n(\mathcal{O}_\mathcal{A}(f) = X) = 1/2$ almost surely for $f$ drawn from the prior.*

*Proof.* Let $\mathbb{X} = \{-1, 0, 1\}$, and consider a GP prior over $f$ such that $f(-1) = f(1) = 0$ and $f(0)$ is a standard normal random variable. Define the algorithm $\mathcal{A}$ such that $\mathcal{O}_\mathcal{A}(f) = \{-1\}$ if $f(0) < 0$ and $\mathcal{O}_\mathcal{A}(f) = \{1\}$ otherwise. Clearly, the target set defined by $\mathcal{A}$ is not complement-independent. Moreover, under the PS-BAX algorithm, $x_n$ is always either $-1$ or $1$. Since the values of $f$ at these points are known, the posterior distribution over $f$ at any iteration $n$ remains equal to the prior. Therefore, it follows that $\mathbf{P}_n(\mathcal{O}_\mathcal{A}(f) = \{-1\}) = \mathbf{P}_n(\mathcal{O}_\mathcal{A}(f) = \{1\}) = 1/2$ for all $n$. $\qquad\square$

## E  Implementation Details

All our algorithms are implemented using BoTorch [33]. Specifically, we use BoTorch's `SingleTaskGP` class with its default settings for all our GP models, except in the top-$k$ GB1 problem, where we employ a deep kernel GP. To fit our GP models, we maximize the marginal log-likelihood. Approximate samples from the posterior on $f$ for both PS-BAX and INFO-BAX are generated using 1000 random Fourier features [52]. Our implementations of PS-BAX and INFO-BAX support automatic gradient computation, enabling continuous optimization when $\mathbb{X}$ is continuous. For INFO-BAX, we use $L = 30$ Monte Carlo samples to estimate the EIG across all problems.

