# OpenReview forum: "Practical Bayesian Algorithm Execution via Posterior Sampling"
_NeurIPS.cc/2024/Conference — NeurIPS 2024 poster_

### Official Review · Reviewer_BTqx · 2024-07-01

**Soundness:** 4
**Presentation:** 4
**Contribution:** 4
**Rating:** 7
**Confidence:** 3

**Summary:**

This paper proposes a new method within the Bayesian algorithm execution framework, which enables finding a target set of points in a highly efficient manner.

**Strengths:**

The problem tackled in this paper is highly important. The proposed method is sound and shows strong improvements vs. relevant benchmarks, in particular with respect to speed and sometimes also in terms of accuracy.

**Weaknesses:**

I don’t see any noteworthy weaknesses, but I am also not an expert in BO and related tasks (only about Bayesian inference more generally).

**Questions:**

The word “posterior sampling” seems to mean something very specific in this paper. However, in the Bayesian literature, this term is used for a much wider set of problems, which may be confusing to some readers of this paper. I think the mention the alternative term “Thompson sampling”. Could this be a good alternative name to use in relevant places?

**Limitations:**

I think the authors could discuss more limitations of this method. Are there any major limitations beyond the fact that the method only works for base algorithms that have a set of points as target?

---

> ### Author Rebuttal · Authors · 2024-08-07
>
> Dear Reviewer BTqx,
>
> We sincerely appreciate your feedback and positive evaluation of our work. We have addressed your comments below and are ready to discuss any new questions or concerns that may arise during the discussion period. Additionally, given your positive assessment, we would deeply appreciate your support in championing our paper during the discussion period. Thank you in advance for your help!
>
> **Q1.** *"The word 'posterior sampling' seems to mean something very specific in this paper. However, in the Bayesian literature, this term is used for a much wider set of problems… the alternative term 'Thompson sampling.' Could this be a good alternative name to use in relevant places?"*
>
> **A1.** Thank you for pointing this out. We initially chose "posterior sampling" instead of "Thompson sampling" because we thought the latter might be too closely associated with optimization settings, whereas our work addresses a broader class of problems. However, in hindsight, we agree that "posterior sampling" may be too broad. We will revise our terminology accordingly in the final version.
>
> **Q2.** *"I think the authors could discuss more limitations of this method. Are there any major limitations beyond the fact that the method only works for base algorithms that have a set of points as target?"*
>
> **A2.** Our approach inherits several limitations from Bayesian optimization. Specifically, the good performance of PS-BAX depends on having a probabilistic model with reasonable predictive capabilities, which can be challenging in some applications. Since this limitation also applies to INFO-BAX and most probabilistic numerical methods, we did not initially highlight it. However, we will add a discussion on this in the revised manuscript. Additionally, as mentioned in Section 3, there is room for more sophisticated theoretical analyses of PS-BAX and INFO-BAX. While we see this as a future research direction enabled by our work rather than a limitation, we acknowledge the importance of developing novel theoretical tools to further analyze these methods.

---

> > ### Comment · Reviewer_BTqx · 2024-08-12
> >
> > Thanks. I will keep my (positive) score.

---

> > > ### Author Response · Authors · 2024-08-13
> > >
> > > Dear Reviewer BTqx,
> > >
> > > Thank you once again for your thoughtful feedback and support of our work.
> > >
> > > Sincerely,
> > >
> > > The Authors

---

### Official Review · Reviewer_Ta95 · 2024-07-11

**Soundness:** 3
**Presentation:** 3
**Contribution:** 2
**Rating:** 5
**Confidence:** 2

**Summary:**

This paper introduces a scalable posterior sampling algorithm (aka PS-BAX) in the framework of Bayesian algorithm execution (BAX). Its fundamental basis is built on a key observation that the property of interest for many tasks is a target set of points, which for many scenarios such as standard bayesian optimization and level set estimation is usually the subset of the domain of the function. The authors propose PS-BAX with two key steps at each iteration: first the algorithm computes the target set using a sampled function from a posterior of Gaussian process; then it chooses the point with highest posterior variance (or entropy). In comparison to prior work (mainly INFO-BAX) based on expected information gain (EIG), PS-BAX shows superior computational efficiency. The authors have also established the asymptotic consistency of PS-BAX assuming the target set is stable.

**Strengths:**

The paper is well-written and the algorithm is validated through extensive and broad classes of experiments with comparisons to other methods.

**Weaknesses:**

This paper is an incremental work on INFO-BAX, which provides an alternative method to get around the expensive EIG computations by limiting the optimization to a target set per iteration. The posterior sampling is intuitive, however, it lacks rigorous development on its connection to INFO-BAX.

**Questions:**

1. The authors claim that PS-BAX is scalable to high dimensional problems. The experiments contain examples only up to 10 dimensions. In the gene embedding experiment, the authors applied PCA to reduce the data to 5 dimensions. I suspect the Gaussian process approximations might be problematic in high dimensions. Can you justify the feasibility of applying PS-BAX to high dimensional data in numerical experiments?

2. INFO-BAX optimizes the mutual information over the entire domain X, whereas PS-BAX optimizes over a sampled target set X_n which is a subset of X. At each iteration (especially early iterations), one can imagine INFO-BAX should select a point that’s more informative on learning the task as it’s over the entire space. However, for some experiments (e.g. Figure 4), PS-BAX performs much better than INFO-BAX for early iterations. Can you provide more intuition on such phenomena?

3. The authors also claim that the PS-BAX method is easily parallelizable as one of the attractive contributions. However, as described in the main algorithm, PS-BAX is based on sequential computation of the posterior for the Gaussian process given samples selected from earlier iterations. Can you provide more details on how it can be parallelizable?

**Limitations:**

My major concern is that within the BAX framework minimizing the number of evaluations is of primary importance. However, PS-BAX seems to have a lower efficiency picking up points as compared to the INFO-BAX method given a limited number of iterations. The study of non-asymptotic results seems to be important to provide practical guidance as compared to asymptotic results.

---

> ### Author Rebuttal · Authors · 2024-08-07
>
> Dear Reviewer Ta95,
>
> We thank you for your feedback and questions. We are glad that you found our paper "well-written" and our empirical evaluation "extensive." We hope to address your concerns in the following clarifications.
>
> **Q1.** *"This paper is an incremental work on INFO-BAX… it lacks rigorous development on its connection to INFO-BAX"*
>
> **A1.** We reiterate that PS-BAX offers superior computational efficiency and empirical performance compared to the state-of-the-art INFO-BAX. Furthermore, unlike INFO-BAX, PS-BAX enjoys a provable convergence guarantee. These facts alone demonstrate that our work is a substantial advancement.
>
> Additionally, though both policies use posterior samples of the target set, the design principles of INFO-BAX and PS-BAX are fundamentally different. INFO-BAX uses them to approximate the expected information gain (EIG) over the target set and then chooses the point maximizing this quantity. In contrast, PS-BAX draws a single posterior sample to identify a region likely to be the target set and then chooses the point with the highest posterior uncertainty within this set. While PS-BAX’s second step is also related to the notion of entropy, the EIG calculation required by INFO-BAX is generally much more computationally demanding, as discussed in Appendix B. We hope this further clarifies the algorithmic difference and highlights the novelty of our work.
>
> **Q2.** *"The authors claim that PS-BAX is scalable to high dimensional problems… I suspect the Gaussian process (GP) approximations might be problematic…"*
>
> **A2.** As noted, GP models struggle in high-dimensional settings. However, this is a limitation of GPs - PS-BAX can easily be paired with other probabilistic models. To demonstrate this and alleviate concerns, we conducted an additional experiment involving a top-K protein optimization task with an 80-dimensional input space using the publicly available GB1 dataset. Instead of a traditional GP, in this experiment, we use a deep kernel GP (Wilson et al. 2015), which scales better to higher-dimensional settings. As shown in Figure 1 of the PDF attached to our rebuttal’s overall response, PS-BAX and INFO-BAX perform similarly (both outperforming Random significantly). However, like in other problems, PS-BAX is much faster to compute.
>
> Indeed, in our paper, we imply that PS-BAX scales better to high-dimensional settings than INFO-BAX. Here, we referred specifically to the computational cost of PS-BAX and INFO-BAX in problems of moderate-to-high dimension. Maximizing the EIG, as required by INFO-BAX, becomes prohibitively expensive in such settings. For instance, in the Ackley 10-D problem, each iteration of INFO-BAX takes over 6 minutes. In contrast, PS-BAX, with its lightweight computation, remains feasible in moderate dimensions (only 15 seconds for Ackley 10-D).
>
> **Q3.** *"...for some experiments (e.g. Figure 4), PS-BAX performs much better than INFO-BAX for early iterations."*
>
> **A3.** The superior performance of PS-BAX over INFO-BAX is due to two key reasons.  First, PS-BAX inherits posterior sampling’s strong exploration capabilities due to the stochastic nature of its sampling choices, allowing it to escape incorrect posterior beliefs of the target set’s identity. This is particularly true in challenging problems, where uncertainty remains significant throughout the experimentation loop. In contrast, INFO-BAX myopically tries to minimize the target set’s posterior uncertainty as much as possible at each iteration, causing it to get stuck in a wrong belief of the true target’s identity. This behavior is observed in Figure 3  (corresponding to the performance plot shown in Figure 4), where INFO-BAX fails to find a portion of the target set due to its lack of exploration. The second reason is computational. As discussed above, maximizing the EIG over high-dimensional spaces is challenging, meaning that despite significant computational efforts, INFO-BAX is potentially choosing to evaluate a local maximum of the EIG instead of the global maximum, thus deteriorating its performance.
>
> **Q4.** *"The authors claim that PS-BAX is easily parallelizable…"*
> A4. We apologize for any confusion. By "parallelizable," we mean that our algorithm can be generalized to the "parallel" or "batch" setting, where multiple points are selected at each iteration (Kandasamy et al. 2018). This generalization follows the approach of Kandasamy et al. (2018). Specifically, to select $q$ points at each iteration, we draw $q$ independent samples from the target set and then select the subset of $q$ points with the highest entropy. To demonstrate our algorithm's empirical performance in the batch setting, we include results for two test problems analyzing PS-BAX under three different batch sizes in Figures 2 and 3 of the PDF attached to our rebuttal’s overall response.
>
> **Q5.** *"My major concern is that… PS-BAX seems to have a lower efficiency.. compared to INFO-BAX ... The study of non-asymptotic results seems to be important…"*
>
> **A5.** We kindly ask the reviewer to consider that our empirical evaluation clearly shows the opposite: PS-BAX is more efficient than INFO-BAX, especially in challenging problems. In contrast, there is no evidence, theoretical or empirical, to substantiate that INFO-BAX is more efficient than PS-BAX. We agree that non-asymptotic results could provide guidance. However, this does not diminish the contributions of our work, which provides an algorithm with faster computation, significantly better empirical performance, and a convergence guarantee.
>
> **References**
>
> Kandasamy, K., Krishnamurthy, A., Schneider, J., & Póczos, B. (2018). Parallelised Bayesian optimisation via Thompson sampling. In International Conference on Artificial Intelligence and Statistics.
>
> Wilson, A. G., Hu, Z., Salakhutdinov, R., & Xing, E. P. (2016). Deep kernel learning. In International Conference on Artificial Intelligence and Statistics.

---

> > ### Comment · Reviewer_Ta95 · 2024-08-13
> >
> > Thanks for the response! The batch-based experiments make sense to me. I have raised the score to 5.

---

> > > ### Author Response · Authors · 2024-08-14
> > >
> > > Dear Reviewer Ta95,
> > >
> > > We are glad that our response addressed your concerns, and we sincerely thank you for raising your score.
> > >
> > > Thank you again for your valuable feedback and support of our work.
> > >
> > > Sincerely,
> > >
> > > The Authors

---

### Official Review · Reviewer_FDTf · 2024-07-12

**Soundness:** 1
**Presentation:** 3
**Contribution:** 2
**Rating:** 5
**Confidence:** 4

**Summary:**

This paper utilizes the idea of Bayesian sampling in the Bayesian algorithm execution (BAX) framework. As claimed by the authors, this is the first extension of posterior sampling beyond the optimization setting. The idea is very simple and natural and the performance seems reasonable. While I am not familiar with the BAX literature, it seems that the theoretical justification is incorrect.

**Strengths:**

Overall, the paper is well structured and well written.

**Weaknesses:**

In terms of computation cost, if a Gaussian process is used, then the computation cost of posterior variance $\sigma_n(x)$ involves the inversion of a high dimensional matrix. Can you please comment on it?

In terms of the theorem 1:
The notation is very confusing. On the LHS ($P_n(X=\mathcal{O_ A}(f))$), $f$ is random with probability measure $P_n$. On the RHS ($1(X=\mathcal{O_ A}(f))$), $f$ becomes fixed. I suppose the author means that on the LHS, $f$ refers to the random sample, and on the RHS, $f$ refers to the true function.

The proof of theorem 1 doesn't involve the prior specification or the true $f$ function, which seems incorrect.
For instance, (1) if the prior distribution is a Dirac measure, then the posterior is also a Dirac measure, and the algorithm never works as the sampling repeatedly returns the same $f_n$. But it won't affect theorem 1? (2) if $f$ contains a flat region of global optimal and $\mathcal A$ aims to find optimums, (i.e., $\mathcal{O_ A}(f)$ is an uncountable set) and the Gaussian process is used,  then with probability 1, $\mathcal{O_ A}(\tilde f_n)$ contains only one element for any $n$. The claimed convergence clearly doesn't hold.

Theorem 2 claims that consistency fails in general, and Theorem 1 claims that consistency holds when we have the stable assumption. This implies that some restriction is necessary, and the stable assumption is a sufficient assumption, not a necessary assumption.

**Questions:**

Please see weakness.

**Limitations:**

The checklist mentions that it discusses the limitation in section 4.3, but I cannot find it.

---

> ### Author Rebuttal · Authors · 2024-08-07
>
> Dear Reviewer FDTf,
>
> We sincerely thank you for your comments and questions. We are glad that you found our paper "well structured and well written." You expressed concerns related to the notation and clarity of our theoretical results. We hope our response below addresses these concerns. Since no other major issues were raised, we kindly ask you to consider raising your rating.
>
> **Q1.** *"In terms of Theorem 1: The notation is very confusing. On the LHS…"*
>
> **A1.** Our statement of Theorem 1 is indeed mathematically correct and we hope to clarify any misunderstandings here. Under a Bayesian perspective, $f$ is a random function with a given prior probability distribution. On both sides of the equation
> $\lim\_{n\rightarrow\infty}\mathbf{P}\_n(X = \mathcal{O}\_{\mathcal{A}}(f)) = \mathbf{1}\set{X = \mathcal{O}\_{\mathcal{A}}(f)}$,
> $f$ refers to the same random function. Furthermore, we note that $\mathbf{P}\_n$ is a random probability measure due to its dependence on the dataset $\set{(x_i, y_i)}\_{i=1}^n$, which in turn depends on $f$. Thus, the claim that “$\lim\_{n\rightarrow\infty}\mathbf{P}\_n(X = \mathcal{O}\_{\mathcal{A}}(f)) =\mathbf{1}\set{X = \mathcal{O}\_{\mathcal{A}}(f)}$ almost surely” should be interpreted as follows: the sequence of random variables $\set{\mathbf{P}\_n(X = \mathcal{O}\_{\mathcal{A}}(f))}\_{n=1}^\infty$ converges almost surely to the random variable $\mathbf{1}\set{X = \mathcal{O}\_{\mathcal{A}}(f)}$ under the probability measure induced by the prior on $f$. We will clarify this in the revised version of our manuscript.
>
> In measure-theoretic terms, Theorem 1 simply states that the random variable $\mathbf{1}\set{X =  \mathcal{O}\_{\mathcal{A}}(f)}$ is $\mathcal{F}\_\infty$ -measurable, where $\mathcal{F}\_\infty$ is the sigma-algebra generated by the sequence of observations $\set{(x_n, y_n)}\_{n=1}^\infty$. Intuitively, this means that given the data $\set{(x_n, y_n)}\_{n=1}^\infty$, we can perfectly determine if the statement “ $\mathcal{O}\_{\mathcal{A}}(f)$ is equal to $X$” is true for any $X$. In particular, the following is a direct corollary of Theorem 1.
>
> *Corollary.* Let $\widehat{X}\_n = \arg\max\_{X\subset\mathcal{X}} \mathbf{P}\_n(X =  \mathcal{O}\_{\mathcal{A}}(f))$ be the time-$n$ maximum a posteriori estimator of $\mathcal{O}_{\mathcal{A}}(f)$. Then, $\widehat{X}\_n$ converges to $\mathcal{O}\_{\mathcal{A}}(f)$ almost surely.
>
> **Q2.** "The proof of Theorem 1 doesn't involve the prior specification or the true $f$ function…"
>
> **A2.** We apologize for the confusion caused by our statement of Theorem 1. As discussed in A1, we follow a Bayesian perspective where $f$ is drawn from the prior distribution used by our algorithm. Our result assumes that the prior is well-specified. Such results are typical in the literature (see, e.g., Russo and Van Roy 2014, Bect et al. 2016, and Astudillo and Frazier 2021) and can be seen as the Bayesian counterpart of frequentist results, which typically assume that $f$ lies in the reproducing kernel Hilbert space of a user-specified kernel. Consistency in the case where the prior over $f$ is a Dirac measure does hold. Regarding the example where $f$ "contains a flat region of global optimum and A aims to find optima… and the Gaussian process is used," we agree that consistency does not hold because the prior is not well-specified in this case. We will revise our statement of Theorem 1 to clarify that $f$ is assumed to be drawn from the prior distribution to avoid this confusion in the future.
>
> **Q3.** "...the stable assumption is a sufficient assumption, not a necessary assumption."
>
> **A3.** We thank the reviewer for pointing this out. We will revise our usage of the word "necessary" in Line 168 to avoid this confusion. As noted, Theorem 1 shows that stability is a sufficient condition for asymptotic consistency, and Theorem 2 shows that asymptotic consistency cannot be guaranteed without stability.
>
> **Q4.** "...the computation cost of posterior variance $\sigma_n(x)$ involves the inversion of a high-dimensional matrix"
>
> **A4.** The computation of the posterior variance scales as $n^3$, where $n$ is the number of training points. While this is seen as a computational bottleneck in other applications involving Gaussian processes, it is not the case in Bayesian optimization and Bayesian algorithm execution tasks, where the maximum number of evaluated points rarely surpasses 1000 due to the assumed high cost of such evaluations. In contrast, the computation of the expected information gain required by INFO-BAX scales as $(n+m)^3$, where $m$ is the size of the target set, also due to the need to compute the inverse of a matrix. Even if $n$ is small, $m$ can be very large in real-world applications such as shortest-path problems and level-set estimation tasks, making INFO-BAX extremely computationally demanding.
>
> **Q5.** "The checklist mentions that it discusses the limitation in section 4.3…"
>
> **A5.** We apologize for the confusion. We meant Section 3 instead of Section 4.3. Additionally, in Section 2 (Lines 77-78), we point out that the range of problems that can be tackled with our PS-BAX strategy is narrower than those that can be tackled with INFO-BAX.
>
> **References**
>
> Astudillo, R., & Frazier, P. (2021). Bayesian optimization of function networks. Advances in Neural Information Processing Systems, 34, 14463-14475.
>
> Bect, J., Bachoc, F., & Ginsbourger, D. (2019). A supermartingale approach to Gaussian process based sequential design of experiments. Bernoulli, 25(4A), 2883-2919.
>
> Russo, D., & Van Roy, B. (2014). Learning to optimize via posterior sampling. Mathematics of Operations Research, 39(4), 1221-1243.

---

> > ### Comment · Reviewer_FDTf · 2024-08-12
> >
> > Dear authors,
> >
> > Sorry for my late reply. I have read all the rebuttals, and have a follow-up regarding theorem 1.
> > On the LHS (i.e., the term $\mathbf P_n(X=O_A(f))$), does $\mathbf P_n$ refer to the posterior measure of $f$? That is, LHS is a RANDOM probability quantity, which depends on random data $(x_i,y_i)$
> > On the RHS, does $f$ still follow the posterior distribution, where the posterior distribution is random as it depends on data $(x_i,y_i)$?
> >
> > If so, then, theorem 1 essentially prove that the posterior distribution of $O_A(f)$ converges to a Dirac measure.
> > In other words, theorem 1 is a posterior concerntration result (posterior distribution concentrates toward certain limit), but NOT a posterior consistency result (posterior distribution concentrates toward truth).

---

> ### Author Response · Authors · 2024-08-13
>
> Dear Reviewer FDTf,
>
> We sincerely thank you for taking the time to read our response and for your follow-up question. Our result is indeed a *Bayesian* posterior consistency result, and we hope our response below clarifies this.
>
> In short, you are right that $\mathbf{P}\_n$ denotes the posterior measure of $f$. You are also right that the LHS denotes a random quantity depending on the random data $\set{(x\_i, y\_i)}\_{i=1}^n$. However, $f$ on the RHS is not a function drawn from the posterior but rather a function drawn from the prior, as we explain below.
>
> Our result should be interpreted as follows. Suppose that a random function $f$ is drawn from a prior distribution $p$. This function will remain fixed throughout the entire data collection process, and the data will emanate from this fixed function in the sense that $y\_i = f(x\_i) + \epsilon\_i$.
>
> Further, suppose that $f$ is unknown to our algorithm, but the prior $p$ is known to our algorithm. Although $f$ is unknown, at each iteration $n$, our algorithm can form a posterior distribution over $f$ using the (correctly-specified) prior $p$ and the sequence of observations $\set{(x\_i, y\_i)}\_{i=1}^n$. Let $p\_n$ denote this posterior and, for any fixed $X\subset\mathcal{X}$, let $q\_n(X)$ denote the probability that $\mathcal{O}\_\mathcal{A}(f) = X$ computed from $p\_n$.
>
> Theorem 1 states that with probability one, for a function $f$ drawn from $p$, $q\_n(X)$ will converge to $1$ if $X=\mathcal{O}\_\mathcal{A}(f)$ and will converge to 0 otherwise. Consequently, our result is indeed a Bayesian posterior consistency result in the sense that the estimated posterior of $\mathcal{O}\_\mathcal{A}(f)$ converges to the Dirac measure of the ground truth.
>
> As a final note, we acknowledge that consistency results are typically stated in terms of convergence of estimators rather than convergence of posterior distributions. We chose to present the latter as it is a stronger result,  but we now realize that such a result is more prone to cause confusion. As discussed in our original response, a direct corollary of Theorem 1 is that the maximum a posteriori estimator of $\mathcal{O}\_\mathcal{A}(f)$ is asymptotically consistent. We will add this corollary to our revised manuscript, along with a discussion of our result inspired by your question.
>
> We hope this explanation addresses your concern. If you have any further questions, please let us know, and we will do our best to respond within the remaining time of the discussion period.
>
> Thank you again for your valuable contribution to improving our work.
>
> Sincerely,
>
> The Authors

---

> > ### Comment · Reviewer_FDTf · 2024-08-13
> >
> > Thanks for the explanation. I finally understand the notation (and the meaning that the prior needs to specify the $f$ well). A better presentation of theorem 1 is needed. Personally, if you say that $f$ follows a prior distribution $p$, and present the LHS as a conditional probability conditioned on data, I could understand it quickly.
> >
> > I would raise my score to 5, as I think the implicit assumption of compact $\mathcal X$ is not favorable.

---

> > > ### Author Response · Authors · 2024-08-13
> > >
> > > Dear Reviewer FDTf,
> > >
> > > We are glad to hear that our response has addressed your concern. We will revise the statement of Theorem 1 based on your feedback to enhance its clarity.
> > >
> > > Thank you again for your valuable feedback and support of our work.
> > >
> > > Sincerely,
> > >
> > > The Authors

---

### Official Review · Reviewer_qVHf · 2024-07-13

**Soundness:** 2
**Presentation:** 3
**Contribution:** 3
**Rating:** 5
**Confidence:** 3

**Summary:**

This work proposes a posterior sampling algorithm for Bayesian algorithm execution, where the goal is to infer the output of an algorithm $O$ applied to an unknown function $f$.  The algorithm is simple to implement and computationally more efficient than previous works based on mutual information maximization.  The authors proved the algorithm is consistent with prior probability 1, and demonstrated empirically that it has comparable and sometimes better sample efficiency than previous work while being faster computationally.

**Strengths:**

- The problem studied is interesting and practically relevant.
- The proposed method is simple, computationally efficient and empirically effective, thereby providing a robust baseline for the problem.
- The paper is mostly well-written.

**Weaknesses:**

- I am uncertain about the correctness of the consistency proof (see question section below).

- A less important point is about a design choice in the algorithm: when the sampled target set has multiple point the algorithm selects the point $x$ with the highest entropy $H[f(x)]$. This does not appear to be a universally optimal choice, for example if we want to estimate a level set $\\{x: f(x)\ge A\\}$ and there exists some $x_0\in \mathcal{X}$ for which the posterior $P(f(x_0)\mid D_n)$ has a very large mean and also a large variance: in such cases $H(f(x_0))$ could be the largest among the sampled level set but there may be little uncertainty that $x_0$ lives in the true level set.  In addition, there may be scenarios where we need to focus on inferring the boundary of the level set instead of reducing the uncertainty about function values in the interior.

**Questions:**

My main question is about the proof of Theroem 1, in particular the claim on L762 that $Z\cap O_A(f)=\emptyset$ "by construction".  I can see this may be true if we assume $P_\infty(O_A(f))$ always assigns positive probability to some set $X\subset \mathcal{X}$, but what if this is not the case, for example, if $P_\infty(O_A(f))$ is a distribution over bounded intervals $\subset \mathbb{R}=\mathcal{X}$ and its marginal distributions for the endpoints are atom-less?  Clearly similar issues can happen if we have (for example) a GP prior with a continuous kernel and consider $P_n$ instead of $P_\infty$.  There should be explanations on why the issue will not happen with $P_\infty$, or why the reasoning of L762 is still valid in the presence of such issues.

**Limitations:**

Yes.

---

> ### Author Rebuttal · Authors · 2024-08-07
>
> Dear Reviewer qVHf,
>
> We sincerely appreciate your feedback and questions. We are glad that you found the problem of study "interesting and practically relevant," our method providing "a robust baseline for the problem," and our paper "well-written." Your major concern relates to the correctness of a specific claim in our proof of Theorem 1. Below, we explain that the claim holds true under an additional assumption (see A1) that we regrettably neglected to include in the statement of Theorem 1. Noticing your high scores for soundness, presentation, and contribution, we hope this clarification will lead you to consider raising your rating. We also address a minor concern related to the optimality of our algorithm in certain settings.
>
> **Q1.** *"My main question is about the proof of Theorem 1, in particular the claim…"*
>
> **A1.** Our proof of Theorem 1 assumes that $\mathcal{X}$ is a finite set. As noted, without this assumption, $\mathbf{P}(X=\mathcal{O}_{\mathcal{A}}(f))$ cannot be guaranteed to be positive for any subset $X \subset \mathcal{X}$. We conjecture that a suitable form of asymptotic consistency for PS-BAX holds even if $\mathcal{X}$ has infinite cardinality. However, we expect such analysis to be significantly more challenging given that techniques commonly used in Bayesian optimization to extend convergence guarantees to continuous input spaces, such as carefully chosen discretizations and Lipschitz assumptions (e.g., Bull et al. 2011 and Srinivas et al. 2012), do not naturally translate to settings where performance cannot be summarized by objective function values. We believe this is the reason why most theoretical results in level set estimation assume that the input space is finite (see, e.g., Gotovos et al. 2013 and Mason et al. 2022). Despite this limitation, our theoretical guarantee still covers a broad range of real-world applications; in particular, the shortest path, top-K, and drug discovery problems explored in our experiments, which are inherently discrete.  We apologize for the confusion and hope this response alleviates any concerns regarding the validity of our theoretical result.
>
> **Q2.** *"A less important point is about a design choice in the algorithm…"*
>
> **A2.** PS-BAX comprises two steps: (1) drawing a sample from the posterior distribution on the target set and (2) selecting a point within the sampled target set. For the second step, we chose to select the point with the highest posterior variance within the sampled target set, which is a simple strategy that can ensure asymptotic consistency, inspired by active learning. Despite its simplicity, this strategy performs well across a broad range of tasks. However, we agree that other strategies tailored to specific applications could potentially improve performance further. We see this as a valuable direction for future research enabled by this work, and the demonstration that posterior sampling can be successfully applied to a broader range of tasks beyond optimization as our primary contribution.
>
> **References**
>
> Gotovos, A., Casati, N., Hitz, G., & Krause, A. (2013). Active learning for level set estimation. In International Joint Conference on Artificial Intelligence.
>
> Mason, B., Jain, L., Mukherjee, S., Camilleri, R., Jamieson, K., & Nowak, R. (2022). Nearly Optimal Algorithms for Level Set Estimation. In International Conference on Artificial Intelligence and Statistics.

---

> ### Comment · Reviewer_qVHf · 2024-08-12
> **quick questions**
>
> Thank you for your response.  I agree my question will be addressed if we assume $\mathcal{X}$ is finite.  Could you point out where the finiteness assumption is made?
>
> I also have a quick question regarding your response to Reviewer fDTf: you said
>
> > Regarding the example where "contains a flat region of global optimum and A aims to find optima… and the Gaussian process is used," we agree that consistency does not hold because the prior is not well-specified in this case.
>
> Could you explain why the prior is not well-specified in this case?  Your subsequent response suggested that the issue will not appear if we restrict to prior-almost every $f$.  But we can have a GP prior that assigns positive (indeed, total) mass to constant functions, in which case the example doesn't seem to be immediately ruled out by the restriction.

---

> ### Author Response · Authors · 2024-08-12
>
> Dear Reviewer qVHf,
>
> Thank you for taking the time to read our response. We address your new questions in detail below.
> > I agree my question will be addressed if we assume $\mathcal{X}$ is finite. Could you point out where the finiteness assumption is made?
>
> As mentioned in our original response (fourth sentence of the first paragraph), we regrettably neglected to explicitly include this assumption. We apologize for this oversight. This assumption will be clearly stated in the revised version of our manuscript.
>
> > I also have a quick question regarding your response to Reviewer fDTf… Your subsequent response suggested that the issue will not appear if we restrict to prior-almost every $f$. But we can have a GP prior that assigns positive (indeed, total) mass to constant functions, in which case the example doesn't seem to be immediately ruled out by the restriction.
>
> We agree that Gaussian priors can place positive mass on constant functions. However, note that Reviewer fDTf mentions an example such that the true function “$f$  contains a flat region of global optima” and then adds that “[if a] Gaussian process is used, then with probability 1, $\mathcal{O}\_\mathcal{A}(\tilde{f}\_n)$ contains only one element…” The combination of these two statements necessarily means that Reviewer fDTf is considering an example where the prior is **not** well-specified. Indeed, if we use a Gaussian process prior such that with probability one $\arg\max\_{x\in\mathcal{X}}f$ is a singleton, this necessarily implies that, with probability one,  $f$ will **not** contain a flat region of global optima.
>
> To address your concern more directly, below we show that if the prior is such that $f$ is constant with probability one and $\mathcal{O}\_\mathcal{A}(f) = \arg\max\_{x\in\mathcal{X}}f$, then asymptotic consistency holds. Although the proof is tautological, we hope it clarifies any misunderstanding. Additionally, we note that this result holds for any choice of sampling decisions $\set{x\_n}\_{n=1}^\infty$. Thus, asymptotic consistency in this specific situation not only holds for PS-BAX, but actually holds for any algorithm.
>
> **Proposition.** *Suppose the prior is such that $f$ is constant with probability one and let $\mathcal{O}\_\mathcal{A}(f) = \arg\max\_{x\in\mathcal{X}}f$, then $\lim\_{n\rightarrow\infty}\mathbf{P}(X=\mathcal{O}\_\mathcal{A}(f)) = \mathbf{1}\set{X = \mathcal{O}\_\mathcal{A}(f)}$ almost surely under the prior for any $X \subset \mathcal{X}$.*
>
> *Proof.* Since $f$ is constant with probability one, we have that $\mathcal{O}\_\mathcal{A}(f) = \mathcal{X}$ with probability one. Consequently, with probability one, $\mathbf{1}\set{X = \mathcal{O}\_\mathcal{A}(f)} = 1$ if $X=\mathcal{X}$ and $\mathbf{1}\set{X = \mathcal{O}\_\mathcal{A}(f)} = 0$ if $X\neq \mathcal{X}$.
>
> Moreover, observe that since the prior only puts mass on constant functions, the same is necessarily true for the posterior. Therefore, $\mathbf{P}\_n(\mathcal{O}\_\mathcal{A}(f) = \mathcal{X})=1$ for all $n$. This also implies that $\mathbf{P}\_n(\mathcal{O}\_\mathcal{A}(f) = X)=0$ for any $X\subset\mathcal{X}$ with $X\neq \mathcal{X}$.
>
> From the above, it follows that, for any $X \subset \mathcal{X}$, $\lim\_{n\rightarrow\infty}\mathbf{P}(X=\mathcal{O}\_\mathcal{A}(f)) = \mathbf{1}\set{X = \mathcal{O}_\mathcal{A}(f)}$ almost surely, as desired. $\square$
>
> We hope this discussion addresses your concerns. Please let us know if any further clarification is needed.
>
> Sincerely,
>
> The Authors

---

> > ### Author Response · Authors · 2024-08-13
> >
> > Dear Reviewer qVHf,
> >
> > As the end of the discussion period approaches, we would greatly appreciate it if you could confirm whether our response has adequately addressed your concerns.
> >
> > We also encourage you to review our recent discussion with Reviewer FDTf, as it addresses similar points and may provide further clarity on the issues you have raised. If any questions remain, please let us know, and we will do our best to respond within the remaining time.
> >
> > If your concerns have been resolved, we kindly ask you to consider raising your rating, as this would reflect the improvements made based on your valuable feedback.
> >
> > Thank you again for your time and efforts in reviewing our manuscript.
> >
> > Sincerely,
> >
> > The Authors

---

> ### Comment · Reviewer_qVHf · 2024-08-13
>
> Thank you for your response. My concerns regarding correctness seem addressed and I will update the score accordingly.
>
> The finiteness assumption is somewhat unfortunate, especially since you only have a consistency result (as opposed to rates of contraction). You mentioned there are technical challenges with discretization. Is there any other scenario where the input space has a very large cardinality (e.g. in graph-related applications) and the consistency proof could still be relevant?

---

> ### Author Response · Authors · 2024-08-14
>
> Dear Reviewer qVHf,
>
> We are glad that our response has addressed your primary concern, and we sincerely thank you for raising your score.
>
> Regarding your question about our consistency result and its finiteness assumption, we would like to emphasize the following points:
>
> - Our result applies to a broad range of critical real-world applications. Indeed, many real-world problems involve large, inherently discrete input spaces. For instance, in the drug discovery application discussed in our work (Section 4.5), the input space consists of a discrete set of 5,000 gene mutations. We also recently introduced a protein engineering application formulated as a top-$k$ optimization problem with an input space of similar size (please see A2 in our response to Reviewer Ta95). Moreover, similar problems in this area can involve input spaces exceeding 100,000 elements. Real-world shortest-path problems also often feature large transportation networks with thousands of nodes.
>
> - Our result is non-trivial even when the input space is small. In practical scenarios, observations are often corrupted by noise, which means that uncertainty regarding the true identity of the target set might persist even if the entire input space is evaluated. Our consistency result, which holds under noisy observations, ensures that PS-BAX can effectively mitigate such uncertainty in the long run.
>
> - The asymptotic consistency of *adaptive* algorithms like PS-BAX is not something that can be taken for granted. In Bayesian optimization contexts, popular algorithms have been shown to lack asymptotic consistency, even in discrete input spaces (see, e.g., Astudilo et al. 2023). The absence of asymptotic consistency often comes with erratic behavior, which may hinder performance in practical scenarios. Thus, the combination of our broad empirical evaluation, demonstrating the strong performance of PS-BAX, with our asymptotic consistency result, provides compelling evidence of PS-BAX’s potential to effectively address real-world challenges.
>
> We hope this discussion clarifies the significance of our asymptotic consistency result.
>
> Thank you again for your valuable feedback and support of our work.
>
> Sincerely,
>
> The Authors
>
> **References**
>
> Astudillo, R., Lin, Z. J., Bakshy, E., & Frazier, P. (2023). qEUBO: A decision-theoretic acquisition function for preferential Bayesian optimization. In International Conference on Artificial Intelligence and Statistics (pp. 1093-1114). PMLR.

---

### Author Rebuttal · Authors · 2024-08-07

Dear reviewers,

We sincerely thank you for your thoughtful comments and questions. We are pleased that you found our paper well-written (qVHf, FDTf, Ta95, BTqx), addressing an interesting and practically relevant problem (qVHf, BTqx), and proposing a sound algorithm (qVHf, FDTf, BTqx) with clear performance improvements demonstrated through extensive empirical evaluation (qVHf, FDTf, Ta95, BTqx).

Our paper's current ratings are as follows:
* **Reviewer BTqx** provided a high appraisal with a rating of 7.
* **Reviewers qVHf and Ta95** both assigned a rating of 4. Despite the moderate ratings, their comments are fairly positive, and their scores for soundness, presentation, and contribution are high. We believe we have addressed their concerns effectively and hope they will consider raising their ratings.
* **Reviewer FDTf** assigned a rating of 3. The only major concern raised was the correctness of our asymptotic consistency result. As we explained in our response, our theoretical result is correct, and the confusion arises from not explicitly stating that $f$ is assumed to be drawn from the prior distribution, a simple improvement we will include in the revised version of our manuscript. Since no other concerns were raised, we kindly ask this reviewer to consider raising their rating.

We have responded to each reviewer’s questions individually, aiming to provide sufficient detail. However, we would like to use this space to globally comment on a few points:

**Contributions of our work:** We introduce PS-BAX, a novel algorithm applicable to a wide range of real-world problems, offering superior computational efficiency and empirical performance compared to the state-of-the-art (INFO-BAX). Additionally, PS-BAX comes with a convergence guarantee (unlike INFO-BAX) and provides novel insights into posterior sampling methods. We believe these contributions are at least on par with the average NeurIPS paper, so we kindly ask reviewers to consider raising their ratings.

**Novelty and algorithm design choices:** PS-BAX is the first application of posterior (Thompson) sampling to the Bayesian algorithm execution setting. Indeed, PS-BAX reduces to posterior sampling when the algorithm is simply selecting the argmax. While there are many ways to extend posterior sampling, we chose a simple yet effective approach: selecting the point with the highest uncertainty among the candidate points returned by the posterior sampling step. Exploring other design choices would be a great direction for future research enabled by our work.

**Theoretical results:** We appreciate Reviewers qVHf and FDTf for requesting more clarification about our theoretical results. As detailed in our individual responses, the issues are minor and can be easily addressed.

Thank you again for your insightful feedback. We look forward to a fruitful and engaging discussion.

Sincerely,

The authors

---

> ### Author Response · Authors · 2024-08-11
> **Follow-up on NeurIPS Submission Discussion**
>
> Dear reviewers,
>
> As the discussion period is underway, we wanted to follow up to ensure our response has addressed your concerns. We believe we have thoroughly addressed the points raised and would greatly appreciate it if you could confirm this. If you have any further questions, please do not hesitate to let us know.
>
> Thank you once again for your thoughtful feedback. We look forward to your response.
>
> Sincerely,
>
> The authors

---

> > ### Comment · Area_Chair_J3i4 · 2024-08-11
> > **Received.**
> >
> > Thanks for your summary of key comments and responses. Every paper will be seriously treated and fully discussed.

---

> > > ### Author Response · Authors · 2024-08-11
> > >
> > > Dear Area Chair,
> > >
> > > Thank you for your response and key contributions to the review process.
> > >
> > > We look forward to any further discussion that could enhance the understanding and impact of our work.
> > >
> > > Sincerely,
> > >
> > > The Authors

---

### Author Response · Authors · 2024-08-14
**Thank You for Your Feedback and Fruitful Discussions**

Dear Reviewers,

As the discussion period comes to a close, we wanted to take a moment to thank you for the active and fruitful discussions. We are glad that our responses effectively addressed your primary concerns, leading to significantly improved ratings of our work (7, 5, 5, 5). Your feedback will be carefully incorporated into our revised manuscript.

Thank you once again for your valuable contributions to the review process and for supporting our work.

Sincerely,

The Authors

---

### Decision · Program_Chairs · 2024-09-25

**Decision:**

Accept (poster)

**Comment:**

This paper considers the Bayesian optimization execution framework, deriving a simple yet effective and scalable posterior sampling algorithm with both theoretical and empirical validation. All reviewers agreed that the proposed algorithm is interesting and valuable. In the meantime, we do received concerns in the soundness of the theory of this paper about the consistency proof and the boundedness assumption. After several rounds of interactions between reviewers and authors, the main concerns have been well addressed and so the consensus was made to accept this paper.

In the final version of this paper, the authors are required to improve the clarity of this paper in
- clearer notation definitions;
- explicitly present the boundedness assumption and discuss its limitation and necessarity of having this assumption
- clarify the consistency result in theorem 1.